# Co-Existence of Iron Oxide Nanoparticles and Manganese Oxide Nanorods as Decoration of Hollow Carbon Spheres for Boosting Electrochemical Performance of Li-Ion Battery

**DOI:** 10.3390/ma14226902

**Published:** 2021-11-15

**Authors:** Karolina Wenelska, Martyna Trukawka, Wojciech Kukulka, Xuecheng Chen, Ewa Mijowska

**Affiliations:** Department of Nanomaterials Physicochemistry, Faculty of Chemical Technology and Engineering, West Pomeranian University of Technology, Szczecin, Piastow Ave. 42, 71-065 Szczecin, Poland; mtrukawka@zut.edu.pl (M.T.); wkukulka@zut.edu.pl (W.K.); xchen@zut.edu.pl (X.C.); emijowska@zut.edu.pl (E.M.)

**Keywords:** battery, metal oxide nanoparticles, carbon spheres

## Abstract

Here, we report that mesoporous hollow carbon spheres (HCS) can be simultaneously functionalized: (i) endohedrally by iron oxide nanoparticle and (ii) egzohedrally by manganese oxide nanorods (Fe_x_O_y_/MnO_2_/HCS). Detailed analysis reveals a high degree of graphitization of HCS structures. The mesoporous nature of carbon is further confirmed by N_2_ sorption/desorption and transmission electron microscopy (TEM) studies. The fabricated molecular heterostructure was tested as the anode material of a lithium-ion battery (LIB). For both metal oxides under study, their mixture stored in HCS yielded a significant increase in electrochemical performance. Its electrochemical response was compared to the HCS decorated with a single component of the respective metal oxide applied as a LIB electrode. The discharge capacity of Fe_x_O_y_/MnO_2_/HCS is 1091 mAhg^−1^ at 5 Ag^–1^, and the corresponding coulombic efficiency (CE) is as high as 98%. Therefore, the addition of MnO_2_ in the form of nanorods allows for boosting the nanocomposite electrochemical performance with respect to the spherical nanoparticles due to better reversible capacity and cycling performance. Thus, the structure has great potential application in the LIB field.

## 1. Introduction

Due to a deteriorating environmental situation (e.g., global warming, and the gradual depletion of oil and hard coal resources), the development of balanced and clean energy resources is extremely important. Clean energy sources such as wind, sun and tidal energy are preferred alternatives to fossil fuels, and they are best utilized with high-efficiency energy storage technologies. Lithium-ion batteries (LIBs)are appealing storage sources due to their unique characteristics, such as their extended life, energy density, low maintenance costs, environmental friendliness and lack of memory effect [1,2]. Although the performance of lithium-ion batteries continues to improve, their energy density, cycle lifetime and productivity remain insufficient for large-scale applications in consumer electronics, and transportation and storage of renewable energy. Much effort has been made to create new electrode materials or to design unique electrode architecture to address the ever-increasing demand for batteries with higher energy density and longer cycle life [3,4,5,6,7]. The electrodes must maintain their integrity across multiple discharge–recharge cycles, which is one of the challenges in their design. Li-alloying agglomeration or the formation of passivation layers, which prohibit the fully reversible injection of Li ions into negative electrodes, reduce the life spans of electrode systems [8,9]. Transition metal oxides (TMOs) have recently found use as electrode material for energy storage devices including LIBs [10]. These materials exhibit a large theoretical specific capacity and high working potential for LIBs (ca. 500–1000 mAhg^−1^) [11]. This is an advantage of TMO application due to the prevention of lithium dendrite formation, which increases safe use.

TMOs such as SnO_2_ [12], Fe_3_O_4_ [13] and MnO_2_ [14] are characterized by a very high theoretical capacity. Furthermore, due to its low cost and low environmental impact, iron oxide is one of the most promising materials. Its disadvantage is that the Li reactivity mechanism of transition metal oxides requires the formation and the decomposition of Li_2_O, accompanying the redox reactions of metal nanoparticles [15]. During this conversion reaction, there is usually a large change in volume, which can cause electrode fracture or deterioration of electrochemical efficiency [16]. In order to address these issues, a carbon material can be used, which can prevent both volume change and aggregation of the nanoparticles [17]. In order to obtain high electrochemical performance, the carbon material should possess advantages such as high surface area and high electronic conductivity. A high surface area provides active sites for the pinning on or embedding of nanoparticles on the carbon surface [18].

Mesoporous carbon materials are characterized by a large specific surface area, which can reduce current density per area unit. Another advantage is the thin walls shorten the diffusion paths. In addition, they are an acceptable electrode material due to low cost, high chemical stability and good processing ability [19,20]. Mesoporous hollow carbon nanospheres fully meet these requirements, and therefore, its application as a carrier of metal oxide nanoparticles appears to be reasonable.

Herein, we present a facile synthesis method of hollow carbon nanospheres (HCS) with two stages of functionalization using transition metal oxides (iron oxides in the form of spherical nanoparticles and rod-like manganese dioxide) as advanced anode material for high-performance LIBs. The prepared Fe_x_O_y_/MnO_2_/HCS nanocomposite combines the advantages of empty carbon spheres, such as stability or adaptation to expanding volumes during the cycle, with a high specific capacity of transition metal oxides.

## 2. Experimental

### 2.1. Synthesis of Mesoporous Core/Shell Structured Silica Spheres (SiO_2_@mSiO_2_ Spheres)

Core/shell silica spheres were used as a hard template to obtain HCS using our previous reported method [12]. In a typical synthesis, ethanol (100 mL, P.P.H. Stanlab, Lublin, Poland), tetraethyl orthosilicate (TEOS, Sigma Aldrich, Beijing, China) (4 mL) and concentrated ammonia (28 wt.%, 6 mL, CHEMPUR, Piekary Slaskie, Poland) were mixed and stirred for 24 h. Then, solid SiO_2_ spheres were dried for further use (SiO_2_). To prepare the SiO_2_@mSiO_2_ spheres, 100 mg of SiO_2_ spheres were dispersed in water (160 mL), ethanol (80 mL) and ammonia solution (28 wt.%, 0.67 mL). Then, 80 mg of the surfactant cetyltrimethylammonium bromide (CTAB, Sigma Aldrich, Beijing, China) and 0.37 mL of TEOS were added. After stirring for 24 h, the product was dried at 80 °C to get SiO_2_@mSiO_2_ spheres.

### 2.2. Synthesis of Hollow Carbon Spheres (HCS)

A chemical vapor deposition (CVD) process was applied to prepared HCS. The core-shell SiO_2_@mSiO_2_ was placed in an alumina boat in a tube furnace (Carbolite GERO, Hope, UK) and a CVD process using C_2_H_4_ as the carbon source occurred for 1 h at 800 °C. After synthesis, SiO_2_@mSiO_2_ covered by carbon was obtained (SiO_2_@mSiO_2__C)_._ To obtain HCS, SiO_2_@mSiO_2__C was treated by hydrofluoric acid (CHEMPUR, Piekary Slaskie, Poland) to remove silica and then washed with water and ethanol several times. Finally, the carbon product was obtained by drying the sample in a vacuum at 80 °C for 12 h.

### 2.3. Functionalization of HCS with Metal Oxides (Fe_x_O_y_/MnO_2_/HCS)

To store metal oxide in hollow carbon spheres, iron (III) nitrate nonahydrate (100 mg, CHEMPUR, Piekary Slaskie, Poland) was dissolved in ethanol; added dropwise to the 100 mg HCS, stirring and heating to 50 °C; and then, placed in a furnace for 2 h at 400 °C. Next, Fe_x_O_y_/HCS (40 mg) was added to a KMnO_4_ (40 mg, CHEMPUR, Piekary Slaskie, Poland) solution in a round bottom flask. The reaction was carried out for 0.5 h at 70 °C. Then, the product was collected by filtration, washed two times with water and ethanol and dried at 80 °C in a vacuum for 24 h.

### 2.4. Characterization

The FEI Tecnai F30 transmission electron microscope (TEM, FEI Corporation, Hillsboro, OR, USA) with a field emission gun operating at 200 kV was used to investigate the morphology of the samples. The elemental mappings were performed via energy-dispersive X-ray spectroscopy (EDX, FEI Corporation, Hillsboro, OR, USA) as the TEM mode. Raman spectra were collected with a Renishaw micro Raman spectrometer (λ = 785 nm, Renishaw, Edinburg, UK). Thermogravimetric analysis (TGA) was carried out on 10 mg samples at a heating rate of 10 °C/min from room temperature to 900 °C under air using a DTA-Q600 SDT (TA Instruments, New Castle, DE, USA). X-ray diffraction (XRD) was conducted on a Philips diffractometer (Malvern, Cambridge, UK) using Cu Kα radiation. The N_2_ adsorption/desorption isotherms were measured on a Micromeritics ASAP 2010M instrument (Micrometrics, Tewkesbury, UK) at liquid nitrogen temperature (77 K). To compute the specific surface area and pore size distribution, the Brunauer–Emmett–Teller (BET) and Barrett–Joyner–Halenda (BJH) methods were used, respectively.

### 2.5. Electrochemical Measurements

The as-prepared Fe_x_O_y_/MnO_2_/HCS nanomaterials were used as electrode materials for LIBs. To prepare the working electrode, active materials, acetylene black (C-NERGY™ SUPER C65, Timcal, Congleton, UK) and PVDF (Solvay Plastics, Warszawa, Poland) were mixed in a weight ratio of 85:10:5. Subsequently, N-methyl-pyrrolidone (NMP, CHEMPUR, Piekary Slaskie, Poland) was added to the powder to form a slurry. The working electrodes were fabricated by coating the slurry onto copper foam (Sigma-Aldrich, Beijing, China) and dried in a vacuum at 80 °C overnight. The testing coin cells were assembled with the working electrode, metallic lithium foil (Sigma-Aldrich, Beijing, China) as a counter electrode, NKK TK4350 film as a separator (Sigma-Aldrich, Beijing, China) and 200 µL LiPF_6_ in 1:1 ethylene carbonate (EC)/dimethyl carbonate (DMC) as the electrolyte (Sigma-Aldrich, Beijing, China). The assembly of the cells was carried out in an argon-filled glovebox (M. Braun Co., Garching, Germany). Electrochemical studies by means of cyclic voltammetry (CV) and galvanostatic cycling with potential limitation (GCPL) were performed. The measurements were executed on a VMP3 multichannel potentiostat (BioLogic, Seyssient-Pariset, France) at room temperature.

## 3. Results and Discussion

The morphology and architecture of the HCS and the HCS with metal oxides were presented (Figure 1) and characterized using TEM analysis. Based on the TEM images (Figure 1), it was found that the cores of solid silica spheres were ~250 nm in diameter. The mesoporous shell has a thickness of ~100 nm. Therefore, the obtained HCS have a size diameter of ~450 nm. The iron oxide nanoparticles were evenly distributed throughout the hollow carbon spheres, and their size was ~15 nm. Microscopic analysis revealed that MnO_2_ was deposited onto HCS in a flat form of thin rods with an irregular surface (Appendix A). Their diameter was ~20 nm while the size of the iron oxide was ~20 nm (Figure 2). To confirm the elemental composition of the sample, EDS mapping was performed. Figure 3 reveals that the Mn and Fe was distributed homogeneously of the carbon shell.

In the next step, XRD patterns acquired from HCS, Fe_x_O_y_/HCS, MnO_2_/HCS and the corresponding hybrid material of Fe_x_O_y_/MnO_2_/HCS were depicted (Figure 4). The black line which corresponds to carbon spheres has significant and broad peaks at 2θ = 24.9° and 42° in response to graphitic carbon planes (002) and (100). Fe_x_O_y_/MnO_2_/HCS exhibits further diffraction peaks which reflect the peaks appearing on the patterns of individual components. The peaks at 2θ = 37.9° and 57.8° correspond to the (211) and (600) MnO_2_ planes, respectively [21]. Based on the obtained pattern, it was found that the sample contains a mixture of iron oxides: Fe_2_O_3_ and Fe_3_O_4_. Diffraction peaks characteristic for Fe_2_O_3_ were identified at 2θ = 24.4°, 35.7°, 44.2°, 49.6° and 62.9°. They are related to the (211), (110), (024) and (214) planes [22]. Two peaks assigned to Fe_3_O_4_ were found, at 2θ = 36.6° (311) and 2θ = 57.8° (511) [23].

First, the BET method was used to investigate the specific surface area of the HCS and the corresponding hybrid composite. The N_2_ adsorption/desorption isotherms are shown in Figure 5A. For HCS, the BET specific surface area is 571 m^2^/g. After functionalization with metal oxides, the surface area decreased to 177 m^2^/g. A reduction in the specific surface area is related to the fact that nanoparticles of iron oxide are both on the surface and in the pores of nanospheres. Additionally, the presence of MnO_2_ on the surface of the nanospheres may cause blockage of the pores [24]. From the adsorption branch, the related mesopore size distribution determined using the BJH approach gives average pore sizes at ~5.53 nm with a predominance of pores of a size at 2.2 nm in the case of the HCS, and 4.97 average pore size and most pores with size 2.57 nm for Fe_x_O_y_/MnO_2_/HCS (Figure 5B). This result suggests the mesoporous nature of the material (Table 1).

The purity of HCS and the quantitative analysis of the nanocomposite was verified by thermogravimetric analysis as shown in Figure 6A. At 540 °C, HCS began to decompose in air. The weight loss accelerated as the temperature rose, until all the carbon spheres were depleted at approximately 735 °C. HCS has an ash percentage of 0 wt.% after combustion at 900 °C, indicating that it is of high purity. In comparison to the pristine HCS, the stability of Fe_x_O_y_/MnO_2_/HCS was weaker. During heating, the nanocomposite burned at 230 °C and ended at 430 °C. About 50 wt.% of the sample decomposed; thus, it can be concluded that the metal oxides are half the mass of the sample. Raman spectroscopy is commonly used to characterize all sp^2^ carbons. The Raman spectra of HCS and the corresponding nanocomposite shows two prominent peaks at 1320 and 1600 cm^−1^ (Figure 6B). The former, named D-band, reveals the defect of the C atomic lattice, while the latter peak, called G-band, represents the stretching vibration of C atom sp^2^ hybrid plane. The relative intensity of D to G provides an indicator for determining the in-plane crystallite size or the amount of disorder in the sample [25,26]. Upon deposition of the metal oxides’ nanoparticles, the relation between the D-band and G-band intensities increases; thus, additional defects formed in the HCS structures.

The prepared materials (Fe_x_O_y_/MnO_2_/HCS) were further evaluated as anode material for Li-ion batteries. The CV curves of the Fe_x_O_y_/MnO_2_/HCS electrode recorded at a scan rate of 0.5 mV s^−1^ are shown in Figure 7A. During the first discharge cycle of the Fe_x_O_y_/MnO_2_/HCS electrode, a strong reduction peak in cathodic scan was observed at ~0.5 V, which is in agreement with the reduction of Mn^2+^ and Fe^3+^ to their metallic states due to the formation of Li_2_O, as illustrated in Equation (1),
FeMnO_3_ + 6Li^+^ + 6e^−^ → Fe + Mn + 3Li_2_O(1)
accompanied by electrolyte decomposition into a solid electrolyte interphase (SEI) layer. During the first anodic [27] charge process of the Fe_x_O_y_/MnO_2_/HCS electrode, only two anodic peaks (1.19 and 2.08 V) can be attributed to the oxidation of metallic Mn and Fe, which are illustrated in Equations (2) and (3):Fe + xLi_2_O − 2xe^−^ → 2 FeO_x_ + 2xLi^+^(2)
Mn + xLi_2_O − 2xe^−^ → 2 MnO_x_ + 2xLi^+^(3)

These two peaks shift to 0.75 and 0.3 V in the following reduction step, indicating improved kinetics. The large peaks at 1.2 and 1.7 V in the charge process are attributable to the two-step oxidation of Mn(0) and Fe(0) to MnO_x_ and FeO_x_, respectively [28]. The two pairs of reduction and oxidation peaks that correspond to the FeO_x_/Fe and MnO_x_/Mn conversions appear to be well overlapped, indicating that the two-step electrochemical reactions are highly reversible.

Figure 7B shows the first, the second and the fifth galvanostatic discharge/charge curves of the Fe_x_O_y_/MnO_2_/HCS between 0.05 and 3.0 V (versus Li/Li^+^). The orange line is attributed to the first cycle charge and discharge capacity of Fe_x_O_y_/MnO_2_/HCS 625 mAhg^−1^ and 1100 mAhg^−1^, respectively. It can be assigned to irreversible effects such as the formation of the SEI layer. After cycling, thin SEI form on the Fe_x_O_y_/MnO_2_/HCS electrode, and additional mesopores form in the hollow structure, resulting in the establishment of linked spaces that are conducive to fast Li^+^ ion and electron transport. The stable SEI layer and hollow space on electrodes can help to stabilize lithiation/de-lithiation and reduce mechanical deterioration caused by discharge volume expansion. In the next step, the charge/discharge profile, with different current densities, was measured. The charge/discharge curves of the Fe_x_O_y_/MnO_2_/HCS composite at various rates are shown in Figure 7C. On both discharge and charge profiles, multiple plateaus can be seen, which are in good agreement with the CV curves. A sequential decay in reversible capacities as the rate increases can be observed. The electrode delivered reversible capacities of 1100, 610, 320, 126, 75 and 42 mAhg^−1^ at current densities from 50 to 1000 mAg^−1^. As shown in Figure 7D, the new anode material exhibits good Li^+^-ion storage capacity and cyclic stability at each current density from 50 to 1000 mAg^–1^. Notably, the fused Fe_x_O_y_/MnO_2_/HCS presented much higher capacities at each stage compared to pristine HCS. Charge-discharge profiles obtained by different applied current densities were used to further study the rate behavior of the Fe_x_O_y_/MnO_2_/HCS electrode. As the rate increased, there was a sequential decrease in reversible capabilities. Figure 7D shows that the electrode delivered reversible capacities of 611, 323, 135, 83 and 46 mAhg^−1^ at current densities from 50 mAg^−1^ to 1000 mAg^−1^. When the current density was reduced to 50 mAg^−1^, the capacity immediately returned to 675 mAhg^−1^. The above results indicate that the hybrid material of the Fe_x_O_y_/MnO_2_/HCS electrode has an excellent rate capability. Additionally, the Fe_x_O_y_/MnO_2_/HCS electrode displays high CE, which in many cycles exceeds 100% (Figure 7E). This can be associated with the irreversible side reaction during the charge process or with an irregular amount of transported Li^+^ ions during charge–discharge processes. In the first case, the side reaction can suggest more capacity is generated than the amount of Li^+^ ions released from the active material. Capacity does not reference the actual storage ability of the electrode; it is estimated based on coulomb counting, e.g., to integrate current vs. time until the cut-off potential is reached. Therefore, if there are any side reactions that consume current without affecting the voltage (e.g., the charge is not actually intercalating to a site in the electrode), then this current is integrated, adding to the capacity. If this occurs in the discharge step, then the Coulombic efficiency (CE) result can be >100%. In the second case, if the Li^+^ ion is less intercalated due to structural interference during the discharge process and the maximum amount of Li-ion is released during the charge process, the CE may exceed 100%. Both scenarios can lead to a gradual degeneration of the structure of the electrode active material and, thus, to a reduced stability (Appendix A).

Hollow carbon spheres decorated by iron and manganese oxides with large surface area and a conductive network enable high accessibility of the active material. Fe_x_O_y_/MnO_2_/HCS composite initially reaches the full theoretical capacity, but degradation effects lead to poor cycle stability. A comparison with the electrochemical performance of other reported HCS composites with Fe_x_O_y,_ or MnO_2,_ shows that this is a promising approach to optimize the cycling stability of the battery. Graphene-wrapped Fe_3_O_4_ synthesized by Zhao et al. [29] shows a better charge/discharge capacity but not satisfying stability. Zhu et al. [30] obtained porous olive-like carbon decorated by Fe_3_O_4_, which presented lower dis- and charge capacities. Wu et al. [31] presented a novel foam-like Fe_3_O_4_/C composite made with gelatin as the carbon source and ferric nitrate as the iron source using a sol-gel process. As a result, the Fe_3_O_4_/C composite electrode demonstrates good rate performance with a reversible capacity of 660 and 580 mAhg^−1^ at 3 and 5 °C, respectively, whereas all composite manganese oxide/carbon presented lower capacity and stability compared to our data [32,33]. Therefore, a combination of two oxides (iron and manganese) significantly improved the capacity of obtained electrodes. The state-of-the-art process provides information about the synergistic effect of such a combination [34]. The synergistic effect of combining such components manifests in improving the reversibility of the electrochemical reaction, buffering large distortions and stresses during discharge-charging processes and preventing aggregation of the active material. This results in high reversible capacity, excellent cycling performance and excellent rate capabilities. The unique MnO_2_ nanorods morphology has been reported as anode material for lithium-ion batteries [35,36,37]. This rod-like morphology is reported to enhance electrochemical properties and was proven in our study. These features, along with the high performance of iron oxides, recommend this hybrid structure as promising for boosting the performance of energy storage devices.

## 4. Conclusions

In summary, the present data demonstrate a facile route for the synthesis of a nanocomposite consisting of HCS and metal oxide hybrid material. The synergistic effect of the components in Fe_x_O_y_/MnO_2_/HCS composite displays the enhancement of electrochemical properties in comparison to pristine HCS. The addition of MnO_2_ nanorods boosts the reversible capacity and cycling performance. The discharge capacity of Fe_x_O_y_/MnO_2_/HCS is 1091 mAhg^−1^ at 5 Ag^−1^, and the corresponding CE is as high as 98%. Therefore, the co-existence of two metal oxides stored in HCS resulted in design of composite that has the potential to be used as anode material for lithium-ion batteries with high cycling stability and boosted performance in comparison to the single metal oxide functionalization approach.

## Data Availability

All datasets generated for this study are included in the article.

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
