# Peer review of "Co-Existence of Iron Oxide Nanoparticles and Manganese Oxide Nanorods as Decoration of Hollow Carbon Spheres for Boosting Electrochemical Performance of Li-Ion Battery"

_materials, 2021, doi:10.3390/ma14226902_

Round 1

Reviewer 1 Report

This manuscript presents a facile synthesis method of hollow carbon nanospheres with two-stages of functionalization with transition metals oxides (iron oxides in the form of spherical nanoparticles and rod-like manganese dioxide) as advanced anode material for high performance LIBs. The prepared FexOy/MnO2@HCS nanocomposite combines the advantages of empty carbon spheres, such as stability or adaptation to expand volume during the cycle, with a high specific capacity of transition metal oxides. The coexistence of two metal oxides stored in HCS allow to design a material with a potential to be used as anode materials of lithium ion battery with high cycling stability and boosted performance in respect to the single metal oxide functionalization what is widely presented in current state of the art. I recommend this manuscript to be published in Materials after minor revision.

  1. Line 33-35, “To meet the ever-increasing demand of batteries with higher energy density and longer cycle life, many efforts have been made recently to develop new electrode materials or design novel structures of electrode materials3, 4, 5.” The following papers are recommended to be cited:

Carbon 16 (2020), 287-298; Nature Communications 12(2021), 4519; Science China Materials 64(2021), 1367-1377;

  1. Line 193, “in Eq. (2), (3):

Fe + xLi2O 2xe →2 FeOx + 2xLi+ (2)

Mn + xLi2O 2xe →2 MnOx + 2xLi+ (3)”

In the chemical reaction equation, the symbol of the electron is recommended to be written as “e-”. Moreover, the oxidation reaction of metallic Mn and Fe are electron loss reactions, and the minus sign seems to be missing in the equation. Therefore, the equation should be as follows:

Fe + xLi2O - 2xe- →2 FeOx + 2xLi+ (2)

Mn + xLi2O - 2xe- →2 MnOx + 2xLi+ (3)

  1. Line 246, “with a reversible capacity of 660 and 580 mAh g-1 at 3C and 5C, respectively.” There should be a space between numbers and units. Please correct it.

  1. In order to demonstrate the stability of FexOy/MnO2@HCS composite electrode materials, the author should test the long cycle stability of the electrode materials.

Reviewer 2 Report

The manuscript is devoted to a very interesting subject - improving the anode material of a lithium-ion battery by the addition of Fe and Mn oxide nanoparticles to hollow carbon spheres (HCS). The information provided in the manuscript is of very high interest, but there are some questions mainly concerning the morphology of the obtained samples.

Line 66. "The prepared FexOy/MnO2@HCS nanocomposite …"

The symbol "@" is usually used to core/shell nanoparticles. And the question arises what authors would like to say: (1) the structure is the HCS core surrounding the FexOy/MnO2 shell, or (2) it is a nanocomposite with the HCS matrix and FexOy/MnO2 distributed in the matrix?

Lines 71-72. "Core/shell silica spheres were used as a hard template to obtain HCS by our previous reported method [17]."

The authors refer to [17] as the paper where they earlier described the method of obtaining HCS nanoparticles. The problem is that it is not the paper written by the authors. Probably it is a misprint and they would like to refer to [18]. In this case it will be better if the authors check all other reference numbers.

Lines 123-124. "The morphology and architecture of the HCS and HCS with metal oxides were characterized using TEM analysis."

To prove the morphology of hollow carbon spheres (HCS) it will be nice to provide a SEM image of a broken sphere to see the hollow core of the nanoparticle.

Line 135." Figure 2. STEM image .."

The red square in the STEM image doesn't correspond to the elemental maps (it is easily seen from the geometry of the image).  It will be better to provide the EDS maps obtained from a whole hollow carbon sphere but not from a fragment.

Lines 129-131. "Figure 2 reveals that the Mn is distributed more in the external part of the carbon shell while the iron oxide is concentrated more in the internal part of the carbon shell."

It is not clear how the authors made the conclusions about the Mn and Fe distribution. For example, Mn distributed uniformly through the HCS volume will provide the same map image. Also it concerns the Fe distribution - in the case of the non-uniform distribution of Fe-O nanoparticles on the surface of the HCS particle the map will actually be the same.

Probably, the authors should discuss the contrast of the HAADF-STEM image to support the models of the Mn and Fe distributions.

Line 131. "However, to confirm HR-TEM has been conducted (Figure 3)."

First of all, it is not clear what should be confirmed by the HRTEM images. And, the second, the HRTEM images of the MnO2 and Fe-O nanoparticles look like as they were obtained from the individual nanoparticles located at a thin carbon film, but not at the HCS nanoparticle.

Lines 131-132.  "Microscopic analysis revealed that MnO2 has been deposited onto HCS in a flat form of thin rods with an irregular surface."

It is not clear how the authors made the conclusion.

Reviewer 3 Report

Dear authors,

1.) I suggest to explain the used abbreviations and notations in the manuscript a little more to readers which may not be affiliated with typical abbreviations in your field of research.

e.g.: What is behind the abbreviation mSiO2? What does the nomenclature SiO2@mSiO2 imply or mean?

I recommend to add a short explanation of the notation used throughout the article.

2.) Page2 Line83: "...to get HCS, SiO2@mSiO2_C was treated...

Is this _C a typo or does the extension _C of mSiO2 mean here? Is this a sample label?

3.) In figure4 the dotted lines for the annotated Bragg-Indices are barely visible on a print-out.
The label for (311) is also attributed to a wrong color (it corresponds to Fe3O4 (deep purple) and not Fe2O3 (green)?

4.) Eq.1) describing the electrochemical reaction is rather incorrect. The reacting species is not FeMnO3 but FeO and MnO2 separately.

It seems to me, that using the sum formula of FeMnO3 is not straightforward, looking at the true chemical composition of the metal oxide HCS, especially as FeMnO3 is a distinct chemical compound ?

Also from the description of the synthesis route of the nanoparticles it is not obvious that this material is formed, rather than a mixture of FeO and MnO2 particles inside the HCS. Perhaps it would be good if the authors would explain the initial state and chemical composition in more detail prior to discussing the redox-reaction behavior.

5.) Page 7 Line229 ff:
...electrode displays high coulombic efficiency, which in many cycles exceeds even 100%(Figure 7E). This can be associated tothe irreversible side reaction during the chargeprocess or with irregular amount of transported Li+ions during charge-discharge processes.

This seems to me very illogic - can you please elaborate more, why the CE is constantly >100%. Perhaps explain the type of the side reaction which you think is responsible in more detail.

Round 2

Reviewer 2 Report

The references [14, 18] should be corrected.

Author Response

Thank you, we corrected references 14, 18.

This manuscript is a resubmission of an earlier submission. The following is a list of the peer review reports and author responses from that submission.